# Automatic detection of Arctic polynyas using hybrid supervised-unsupervised deep learning

Céline Heuzé[1] and Carmen Hau Man Wong[1]

[1]Department of Earth Sciences, University of Gothenburg, Gothenburg, Sweden

**Correspondence:** Céline Heuzé (celine.heuze@gu.se)

**Abstract.** Polynyas are open water or thin-ice areas within the ice pack. They play a crucial role for the Earth system, from deep water to cloud formation, causing large gas exchanges, and acting as hotspots for marine life. Yet their monitoring in the Arctic is challenging because polynya detection is non-trivial, owing to the Arctic's complex geometry. Recently, a labelled dataset was released in which daily winter Arctic sea ice concentration since 1978 was turned into a polynya mask. After oversampling to reduce the class imbalance from 0.1% to 2.5%, we use this labelled dataset to train a Unet autoencoder to detect polynya pixels in daily sea ice concentration images. We further filter out the false positives in the marginal ice zone using an unsupervised Gaussian Mixture Model classifier. False negatives are virtually absent from 2012 onwards, when noise in the labelled dataset is reduced by combining ice concentration and thickness masks. False positives exhibit a significant trend with time and anticorrelation with the Arctic sea ice area. Coupled with our expert assessment of individual images, we argue that most "false positives" are in fact correct, detecting patterns of reduced ice within the changing, more unpredictable ice cover that the rigid traditional methods with fixed thresholds cannot identify. We also successfully apply our trained model to detect polynyas in daily and monthly climate model output at low computing costs. As Arctic sea ice continues to decrease, pushing traditional methods to their limits, we expect such machine-learning methods to become the norm.

## 1 Introduction

Polynyas, openings in the pack ice generally less than 100 000 $km^2$ in the Arctic, play a crucial role for the climate and ecosystem. Winter polynyas in particular, by exposing the warm ocean to the frigid atmosphere, act as "ice factories" (Smith Jr and Barber, 2007), with just a few polynyas being responsible for most of the sea ice production in the Arctic (Tamura and Ohshima, 2011; Preußer et al., 2016). The intense air-sea interaction also induces a strong heat- and moisture- loss to the atmosphere (Morales Maqueda et al., 2004; Zhou et al., 2023) that results in cloud formation locally (Monroe et al., 2021) and even impacts the regional and large-scale atmospheric circulation (Gordon et al., 2007). In the ocean, the brine rejected during ice re-formation in the polynya causes deep water formation (Martin and Cavalieri, 1989; Ohshima et al., 2016). The combined effect of ocean mixing and air-sea exchanges further leads to intense gas exchanges and nutrient upwelling (Else et al., 2013; Marchese et al., 2017), making polynyas hotspots for marine life (Moore et al., 2023; Golledge et al., 2025).

Automatically detecting polynyas around Antarctica is somewhat straightforward due to the zonal distribution of ocean - sea ice - land (e.g. Mohrmann et al., 2021). In the Arctic in contrast, that distribution is way less regular. Just in the Nordic

Seas, to the west sea ice can extend to the southern tip of Greenland while to the east north of Svalbard can be ice-free. The land geometry is also more complex, with polynyas often forming in the lee of the region's many islands (Smith Jr and Barber, 2007). Finally, Arctic polynyas are much smaller than their Antarctic counterparts (Morales Maqueda et al., 2004). The result is that few real pan-Arctic polynya studies exist; instead, researchers tend to focus on a series of regions with recurrent polynyas (e.g. Tamura and Ohshima, 2011; Preußer et al., 2016). In these regions, they then use a fixed threshold in sea ice properties to distinguish polynyas from the pack ice. This threshold is chosen somewhat arbitrarily, ranging from 30% (e.g. Tamura and Ohshima, 2011) to more than 60% (e.g. Monroe et al., 2021) for sea ice concentration and 10 (e.g. Martin et al., 2004) to 30 cm (e.g. Smedsrud et al., 2006) for sea ice thickness. The reason for this wide range of thresholds is that there is no strict definition of a polynya (Smith Jr and Barber, 2007); authors instead choose thresholds adapted to the processes that cause or are caused by polynyas they are most interested in studying. As throughout the Arctic sea ice concentration and thickness are dramatically decreasing with climate change (Stroeve and Notz, 2018; Jahn et al., 2024), it is unclear whether and for how long these thresholds will remain valid. Besides, the more winter observations of the Arctic we gather, the more we understand that even a small local decrease in sea ice is enough to trigger some processes (Hoppe et al., 2024; Loose et al., 2024). There is an urgent need to try novel approaches, for full pan-Arctic automatic detection.

Enter machine learning. As recently reviewed by Bracco et al. (2025), examples abound of applications of machine learning methods to climate science, including in the polar oceans (Sonnewald et al., 2021). The most famous example for Arctic sea ice probably is IceNet (Andersson et al., 2021), a series of Unet networks that can forecast Arctic sea ice extent up to six months in advance. A Unet network consists of two convolutional neural networks (CNN), i.e. neural networks that identify patterns in blocks of an image. The first CNN, named the encoder, "zooms in" on the image, while the second one, the decoder, "zooms back out" so that the result of Unet is a pixel-wise classification. Originally designed for medical applications (Ronneberger et al., 2015), this architecture has proven particularly suited to so-called anomaly detection, i.e. finding a rare event occurrence among a majority of normal data points, with applications ranging from detecting image forgery (Choudhary et al., 2024) or electricity theft (Aslam et al., 2020) to pest infestation in forests (Ye et al., 2022). Unet is a type of supervised classifier, which means that it needs labelled data to learn from. Wong et al. (2025) recently produced such a labelled dataset for winter Arctic polynyas since 1978, at a daily resolution; more details about this dataset are provided in the next section. We here leverage this dataset and use it to train a Unet network to automatically detect Arctic polynyas. Following Liu et al. (2025)'s work on lead detection in the Arctic, we improve our results by combining Unet with a Gaussian Mixture Model (GMM), an unsupervised classification method widely used in climate science (e.g. Jones et al., 2023; Paçal et al., 2023; Poropat et al., 2024).

The objective of this manuscript is dual: First automatically detecting Arctic polynyas in satellite observations using hybrid supervised-unsupervised machine learning, then briefly investigating whether our new method can be applied to climate model data. We present the satellite and model data in section 2, in which we also describe the Unet (section 2.3) and GMM (section 2.4) models. We then evaluate and discuss their application to observations (sections 3.1 and 3.2) and to climate model output (section 3.3). We summarise our findings and make final remarks in the conclusions.

## 2 Data and Model architectures

### 2.1 Observed and modelled sea ice data

The objective of this paper is the automatic detection of polynyas from individual Arctic sea ice concentration (SIC) maps. Our input data are the daily SIC from the National Snow and Ice Data Center (NSIDC). Briefly, the daily SIC is obtained from satellite microwave radiometers processed with the NASA Team algorithm (Cavalieri et al., 1999). We use the daily data in November to April, from 1978 to 2023, at a 25 km resolution. We generated the daily sea ice area and sea ice extent from this time series, using the common threshold of 15% concentration for the sea ice extent and no threshold for the area.

For supervised learning, a labelled dataset is required. Here we use the daily polynya mask produced by Wong et al. (2025), based on the NSDIC SIC described above and hence on the same spatial grid as our input data. Wong et al. (2025) also produced a second mask, based on the Soil Moisture Ocean Salinity (SMOS) and Soil Moisture Active Passive (SMAP) derived sea ice thickness (SIT Paţilea et al., 2019). In brief, Wong et al. (2025) detect the pack ice (SIC > 0% or SIT > 0 cm) using a flood-fill algorithm, seeded in the Atlantic and Pacific oceans. Within the pack ice north of 65°N, they then identify polynyas as any pixel with SIC $\leqslant$ 50% or SIT $\leqslant$ 20 cm. On the masks, the pixel value is 0 if that pixel is not a polynya, and 1 if it is.

We use their SIC mask for training, validation, and testing our model. We use their SIT mask to better analyse our false positives (model says there is a polynya, but the mask says there is none) and false negatives (model says there is no polynya, but the mask says there is) after 2012, i.e. when SIT data became reliable. The two masks are available at a daily resolution but only for the winter months, November - April included. We therefore also limit our analysis to these months. The dataset hence consists of 7300 daily images.

Finally, to demonstrate the wider applicability of our method, we apply it to the global coupled model MIROC6 (Tatebe et al., 2019). We chose this model for two reasons: it provides both daily and monthly sea ice concentration ("siconc"); and the brief polynya analysis of Heuzé et al. (2023) revealed that it features large and small polynyas in the Arctic. The model's nominal horizontal resolution for the sea ice grid is 1/3°, which in the Arctic is approximately 30 km. We used the last 30 years of SIC from its historical run, i.e. 1 January 1985 - 31 December 2014.

### 2.2 Input preparation and oversampling

The original NSIDC grid has 448 by 304 pixels. It extends far south into the open Pacific and Atlantic oceans, and includes many land pixels (orange on Fig. 1). For the unsupervised detection of the open ocean described at the end of this section, we keep the grid as it is. But for most of the work, we instead use a reduced grid with 192 x 192 pixels, chosen so that it includes the entire Arctic Ocean (blue on Fig. 1). On both grids, we set all NaNs caused by the pole hole, land, and the very rare missing data to 1, i.e. SIC = 100%.

We originally wanted to preserve the time dimension, assuming that the model would learn from the day-to-day polynya formation and shape evolution. Unfortunately, polynyas are so-called "rare events" on the images. That is, on each daily image on average polynya pixels covered less than 0.05% of the image, barely rising to 0.1% after image size restriction. After

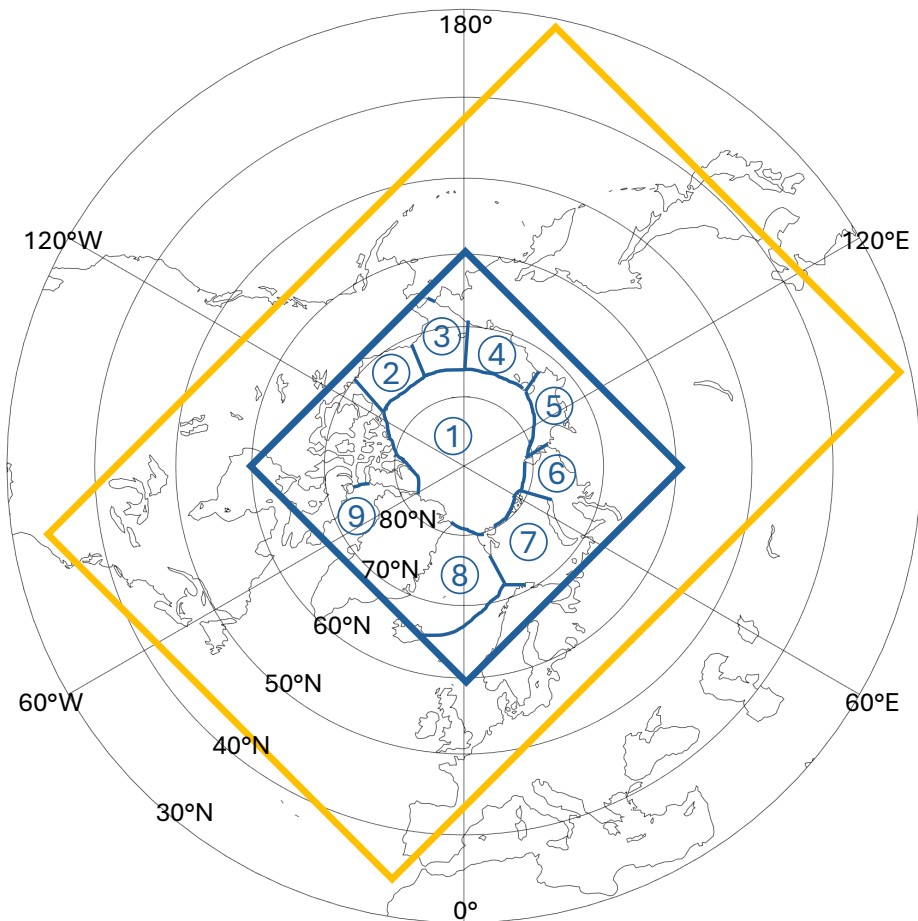

**Figure 1.** The two stereographic grids used in this study: The original 448 x 304 pixel grid from NSIDC used for the unsupervised network is shown in orange; the reduced 192 x 192 pixel grid used for the supervised learning, in blue. For increased readability, all subsequent figures will show only the rectangular/square regions with pixel data, rotated. Blue numbers in a circle indicate the nine Arctic regions, as defined by Meier and Stewart (2023): 1. Central Arctic Ocean; 2. Beaufort Sea; 3. Chukchi Sea; 4. East Siberian Sea; 5. Laptev Sea; 6. Kara Sea; 7. Barents Sea; 8. East Greenland Sea; 9. Baffin Bay.

unsuccessful training, we reluctantly gave up on the time dimension and instead created an oversampled training dataset of 1000 192x192 pixel scenes consisting of a random mix of:

- scenes that combine 500 randomly selected daily images;

- 200 scenes that combine 200 randomly selected daily images;

95 - 400 scenes that combine 100 randomly selected daily images;

- 200 scenes that combine 50 randomly selected daily images;

– and 100 scenes that combine 10 randomly selected daily images.

The polynya mask was set to 1 on the oversampled scene if the grid cell was 1 on at least two of the original images. Visual inspection (not shown) revealed that two images was the minimum required to reduce noise. Where this new, oversampled polynya mask was set to 1, the sea ice concentration was set to the mean sea ice concentration of only the images where the original polynya mask was 1. The sea ice concentration elsewhere was set to the mean of all images elsewhere. Visual inspection (not shown) revealed that the transition polynya/not polynya looks more natural on these oversampled scenes if prior to averaging we set all concentrations lower than 15% to 0.

The resulting oversampled dataset has an average of 2.5% polynya pixels per scene. This is still a large class imbalance, but it is on par with what most anomaly detection algorithms can manage (Krawczyk, 2016; Ghosh et al., 2024).

## 2.3 Supervised binary classification polynya / not polynya with a Unet autoencoder

Using Keras (Chollet and The Keras Team, 2015), we implemented a Unet autoencoder for the supervised classification of our highly imbalanced dataset. See schematic on Fig. 2:

– The encoder is made of two levels with two 2D-convolutional layers of filter size 16 (first level) then 32 (second level) and kernel size of 3 with a relu activation, followed by a 2D maxpooling with a pool size of 2 by 2;

– The bottleneck on the third level is made of two 2D-convolutional layers of filter size 64, also with a kernel size of 3 and a relu activation;

– The decoder is made of two levels of upsampling done by conv2Dtranspose of filter size 32 (second level) then 16 (top level), kernel size of 2, and 2 strides; a concatenation; and two 2D-convolutional layers of filter size 32 (second level) then 16 (top level) and kernel size of 3 with a relu activation. The output layer is a final 2D convolution of filter size 1, kernel size 1, and a sigmoid activation.

Due to the class imbalance, we used a weighted binary cross entropy as loss function and monitored the precision and F1 score. We used an Adam optimiser. The oversampled set was randomly split between 70% for training, 15% for validation and 15% for testing. The best model was chosen based on its performance on the validation set.

We varied the weight from 0 to 100 with an interval of 5, and obtained the lowest number of false positives with a weight of 15 - the number of false negatives was not significantly affected. Similarly, we varied the batch size from 2 to 40, and obtained the fewest false positives with a batch size of 5. For the F1 score, the classifier is so certain of its results that varying the threshold had no impact on the performance; we kept the default threshold of 0.5. For each setting, we trained an ensemble of 100 models.

The final performances, on the validation set, are an F1 score of 0.98 and precision of 0.97. We obtained 438 false negatives and 3 943 false positives, compared to 123 283 total polynya pixels in the oversampled validation set.

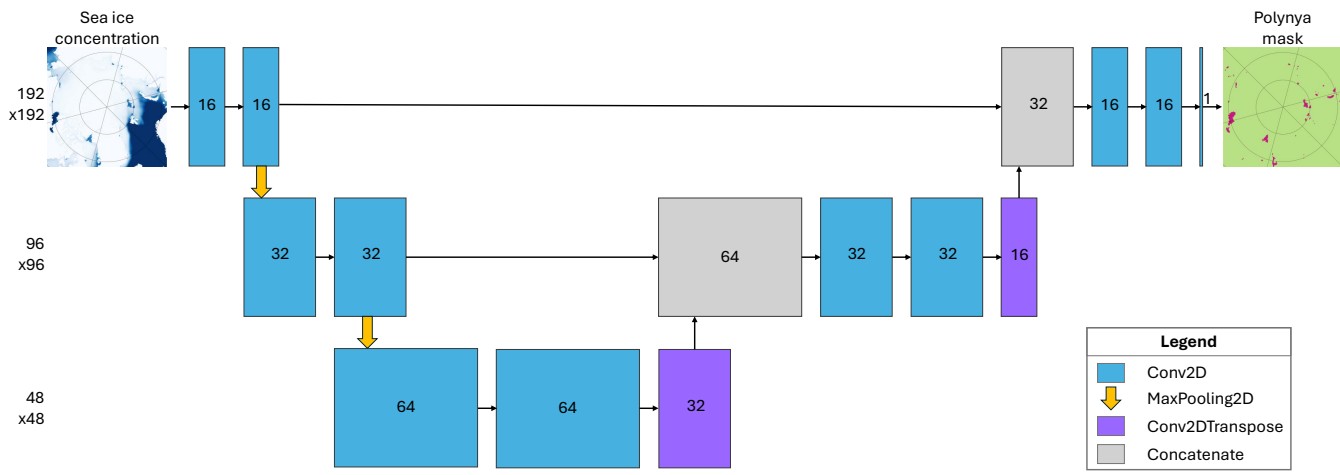

**Figure 2.** Schematic architecture of our Unet implementation. Numbers indicate the output dimensions. See text for the settings of each layer.

## 2.4 Filtering via unsupervised detection of the ocean and MIZ by Gaussian Mixture Models

With our Unet autoencoder, we obtain very few false negatives but a large number of false positives. Rather than modifying the architecture of the network, we filter its daily output using a second, unsupervised machine learning method: clustering using Gaussian Mixture Models. We use the Scikit-learn implementation (Pedregosa et al., 2011), with 3 components, a tolerance of 0.001, up to 200 iterations, and 3 initialisations using kmeans.

The setting that had the most dramatic impact of the performances was the number of components, i.e. how many different classes the model should detect. We varied the number from 3 to the 80 used by Liu et al. (2025). Three was chosen intuitively, assuming the network would split the data into open ocean, land, and pack ice classes. We expected the best performances for 4 or 5 classes, assuming the marginal ice zone (MIZ) would be a class of its own, but saw that for any class number larger than 3, polynyas and MIZ were bundled together in the same class. We therefore used 3 classes.

The input data was the original 448 x 304 pixel dataset, in order to maximise the amount of open ocean pixels the network could learn from. Prior to training, we set the SIC lower than a given threshold to 0, to merge the MIZ and open ocean into one class. We tested thresholds from 0 to 60%, starting from 0 and increasing by 5%, feeding the GMM 2 to 10 daily images at a time, and obtained the best agreement with Wong et al. (2025)'s masks with a threshold of 40% for 2 days. Two days is the minimum the GMM can take, and probably yielded the best results because the MIZ changes very rapidly. The higher the SIC threshold, the better the MIZ was filtered out; however, for any threshold larger than 40% polynyas started being detected as belonging to that open ocean / MIZ class, which is also why we stopped testing at 60%. For the modelled data, 40% yielded satisfactory results, but owing to model biases, 15% was the best compromise to remove the MIZ without removing polynyas as well. When working with monthly modelled data, the GMM was fed two months at a time; for daily modelled data, 2 days, as for the observations.

For each timestep, we ran an ensemble of 10 models. Since class numbers are randomly assigned (see example on Appendix Fig. A1), we identify for each ensemble member the class number corresponding to the middle of the North Atlantic Ocean (median class number in the white band on Appendix Fig. A1, approx 48 - 59N and 47W - 10W). We then mark each pixel as "open ocean or MIZ" if strictly more than 5 of the 10 ensemble members put it in the open ocean class. We chose to detect the open ocean class rather than the pack ice because we noticed while randomly checking individual timesteps that there were occasions where half of the pack ice was detected as land. Only the open ocean / MIZ was consistently its own class. Besides, the pack ice class is noisy, detecting "ice" off the southeast coast of England for example on Fig. A1.

Any pixel that our Unet autoencoder said was a polynya but was found to be in this open ocean / MIZ class was set to 0 or "not a polynya". The effect of this filtering is quantified in the next section.

## 3 Results and Discussion

### 3.1 Detection of Arctic winter polynyas in sea ice concentration observations

After training, we fit the best model to the original 7300 daily sea ice concentration images and compare the model's classification to the SIC polynya mask of Wong et al. (2025). The performances (Table 1, first column) are obviously less impressive than on the validation, oversampled set: We obtain 1.4 million false positive pixels, which is 5 times the number of true positive pixels in the dataset (262 067). After applying the GMM-produced open ocean / MIZ filter (Table 1, second column), the number of false positives is more than halved and the already-low number of false negatives further decreases to just over 10 000.

| Metric | Unet only | + GMM | + SIT mask |
|---|---|---|---|
| False negatives | 12 776 | 10 048 | 6 271 |
| False positives | 1 475 630 | 601 633 | 630 011 |

**Table 1.** Number of false negatives (missed polynya pixels) and false positives (pixels incorrectly identified as a polynya) after the three main steps of the pipeline: Straight out of the supervised classification (Unet, section 2.3); after applying the Gaussian Mixture Models-produced mask (GMM, section 2.4); and after correcting the original sea ice concentration mask by merging it with the sea ice thickness one (SIT). For comparison, there are 107 578 polynya pixels in the dataset after combining the sea ice concentration and thickness masks, and just over 269 million pixels in total (all pixels x all time steps).

A visual inspection of our results made us suspect noise in the masks as the main reason for the number of false negatives. To reduce this noise, we combine the SIC and SIT masks of Wong et al. (2025) from 2012 onwards, i.e. when SIT became widely available in the Arctic. The effect is striking: The number of false negatives drops to near-zero after 2012 (Fig. 3a).

The number of false positives in contrast seems to have an increasing trend with time, even after applying the filter (Fig. 3b). In fact, we do identify a trend significant at 99% of +277 false positive pixels per year, which corresponds to +12 763 pixels over our 46-year time series. Visual inspection of the sea ice maps further suggested that the years with a reduced ice cover

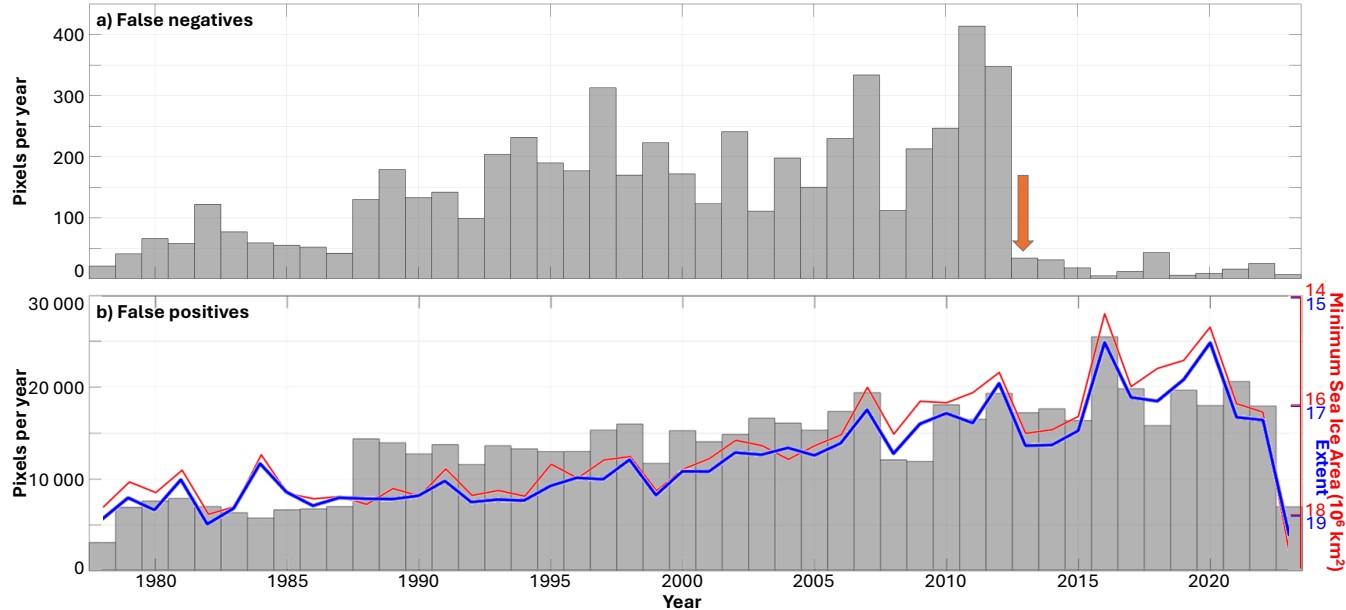

**Figure 3.** a) False negatives and b) false positives for each year, in number of pixels. Note the different y-axes. The orange arrow on a) highlights the beginning of reliable sea ice thickness observations, and therefore the effect of merging the SIC and SIT labelled datasets. The red and blue lines on b) are the minimum sea ice area and extent, respectively (winter months only), flipped vertically to highlight the correlation. Note that the 2023 data stop in April, hence the higher sea ice areas and extents.

coincided with the years with most false positives. This is confirmed by a correlation analysis: with 99% significance, the daily number of false positives and the daily sea ice area (SIA) / sea ice extent (SIE) are strongly anticorrelated (R=-0.76 / -0.71), i.e. the less sea ice, the more false positives. Fitting a regression line through this result, this gives -50 / -57 false positive pixels per million $\text{km}^2$ of Arctic sea ice. As pictured on Fig. 3b, the correlation holds with yearly values too, reaching -0.78 / -0.80 between the yearly false positives and the yearly (winter only) minimum SIA / SIE.

Could this trend rather mean that the GMM is becoming less efficient at filtering out the MIZ? To investigate this, we split the false positive series in the distinct Arctic regions as defined on Fig. 1, and in two month periods that cover the possible processes at play: potential late freeze-up in November-December, near-certain consolidated pack ice in January-February, and potential early-melt in March-April (Fig. 4). The freeze-up period Nov-Dec dominates the distribution and has the largest trend (Appendix Table A1) in three regions: the Beaufort, Chukchi, and Kara seas. In these three regions, admittedly, the false positives may be the result of unfiltered MIZ. Two more regions are dominated by the potential early-melt period March-April: the East Siberian and Laptev seas. Despite the increased seasonality of sea ice in these regions (e.g. Onarheim et al., 2018), such a widespread early-melt seems unlikely. These regions are known hotspots of polynya activity (e.g. Preußer et al., 2016); the false positives may rather indicate a lengthening of the polynya season, a result also found by Wong et al. (2025). Finally, the decreasing trends in false positive pixels (Appendix Table A1) in the Barents and East Greenland Sea will be addressed in

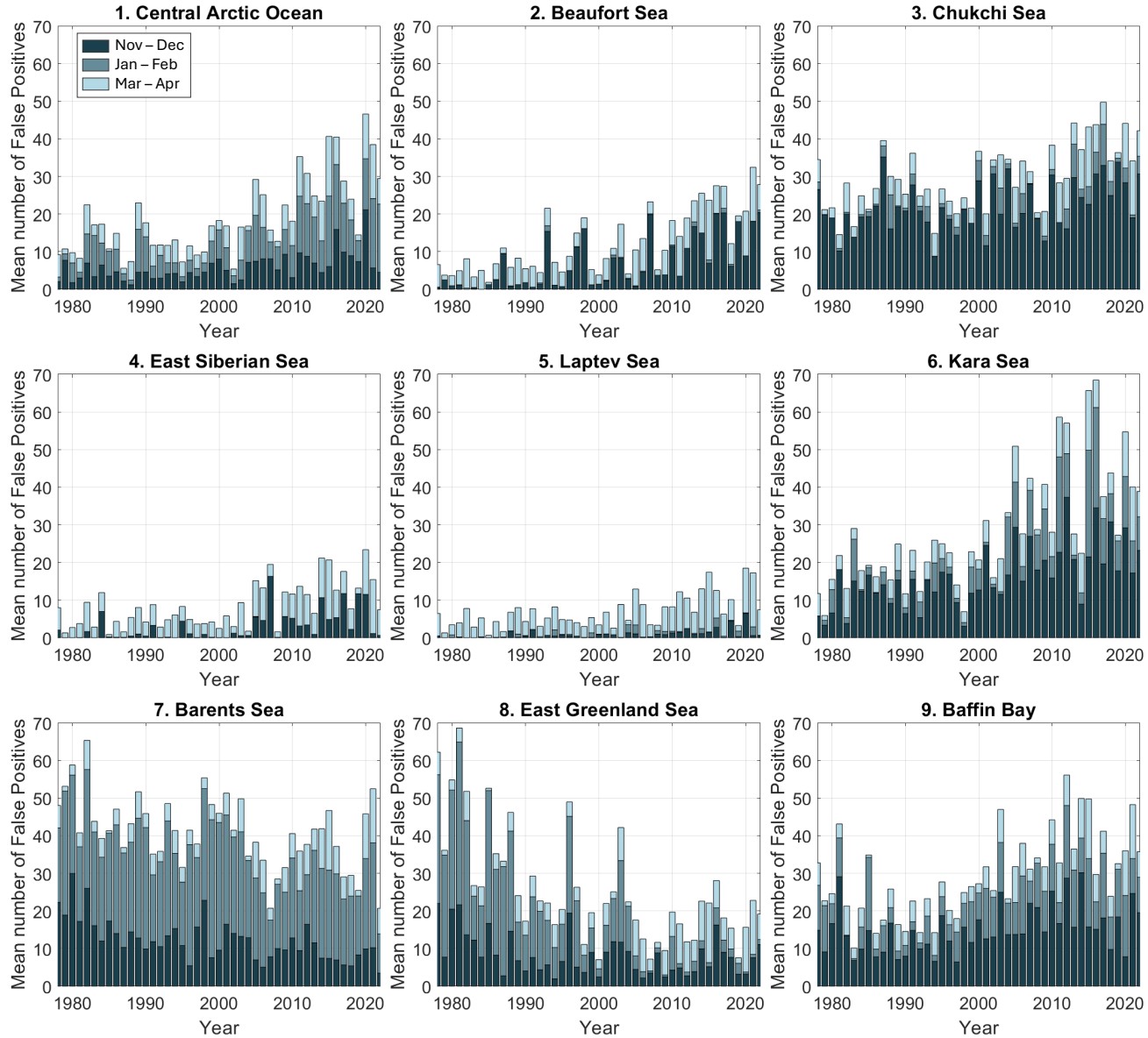

**Figure 4.** Mean number of false positive pixels per two-month period: freeze-up in November-December (darkest), consolidated ice in January - February (greyish blue), and potential early melt in March - April (lightest), for each of the nine regions as defined in the NSIDC Arctic regional mask (Meier and Stewart, 2023, same numbers as on Fig. 1).

the next subsection, and the largest number of false positives for most regions in the consolidated ice January-February period (Fig. 4) is the topic of the next paragraph.

A visual inspection also shows that many of these false positives seem correct. We show two examples on Fig. 5, and first discuss the areas circled in green. For both dates, although our method (magenta) correctly identifies the same polynyas as in the labelled masked (green, hereafter referred to as the traditional method), the polynyas detected by our method are larger. When visually comparing to the original sea ice concentration data, we agree more with our method than with the traditional one. The area circled in blue on Fig. 5b is a special case: it was also detected as a polynya by Wong et al. (2025) but has been removed during their post-processing since it is in a river. Hence technically, our two methods agree there as well. Moving to the areas where the traditional method found no polynya at all, circled in black on Fig. 5, our visual inspection reveals that these could qualify as polynyas, if one defines a polynya as an area where sea ice concentration is significantly less than the surrounding ice. We do acknowledge that our method is not perfect: the MIZ mask filters out some polynya pixels whose surrounding ice also has relatively low ice concentration, while in some complex MIZ regions, the MIZ mask fails to filter out some false positives. Nonetheless, our results highlight the limitation of the traditional way of detecting polynyas using a fixed threshold, and probably explains why so many different sea ice concentration thresholds are used in the literature (30 to more than 60% in e.g. Tamura and Ohshima, 2011; Campbell et al., 2019; Mohrmann et al., 2021; Zhou et al., 2022; Bennett et al., 2024). Since convolutional neural networks detect shapes and gradients, they are better suited to this pattern recognition exercise. The increase of not-that-false positives with increasing climate change also suggests that as sea ice has entered a new regime (Stroeve and Notz, 2018; Sumata et al., 2023) and become more variable (Dörr et al., 2023), polynyas become harder to detect with traditional methods whereas the machine learning methods are not affected: the processes causing polynyas are still happening, albeit from a shifted baseline that may fall below the fixed thresholds.

Admittedly, Arctic polynyas are more often detected using sea ice thickness since they can be covered with a thin layer of high concentration ice (e.g. Martin et al., 2004; Smedsrud et al., 2006; Tamura and Ohshima, 2011; Adams et al., 2013; Preußer et al., 2016; Ren et al., 2022). Since adequate sea ice thickness data are only available since 2010s and that we have to combine images to create the oversampled training dataset, we could not train our model on SIT without grossly overfitting it. It should however be possible to perform transfer learning with minimum retraining, by min-max normalising the SIT so that it too has values between 0 and 1, and most crucially finding an adequate threshold for Unet and the GMM. Similarly, we used a low spatial resolution SIC product for its time coverage. Transferring to higher resolution products should be straightforward and only require that one changes the size of the Unet input layer; the GMM can be directly applied as is, although computing times increase rapidly with resolution. It should also be easy to modify the input layer so that it takes the brightness temperature or the ratio of several frequencies rather than the derived SIC, to obtain results more directly comparable to e.g. Markus and Burns (1995). Finally, we limited our study to winter polynyas because the labelled data were only available for the months November to April, but we see no reason why our method would not function to detect summer polynyas from SIC as well, potentially after minor retraining or threshold adjustment for the GMM.

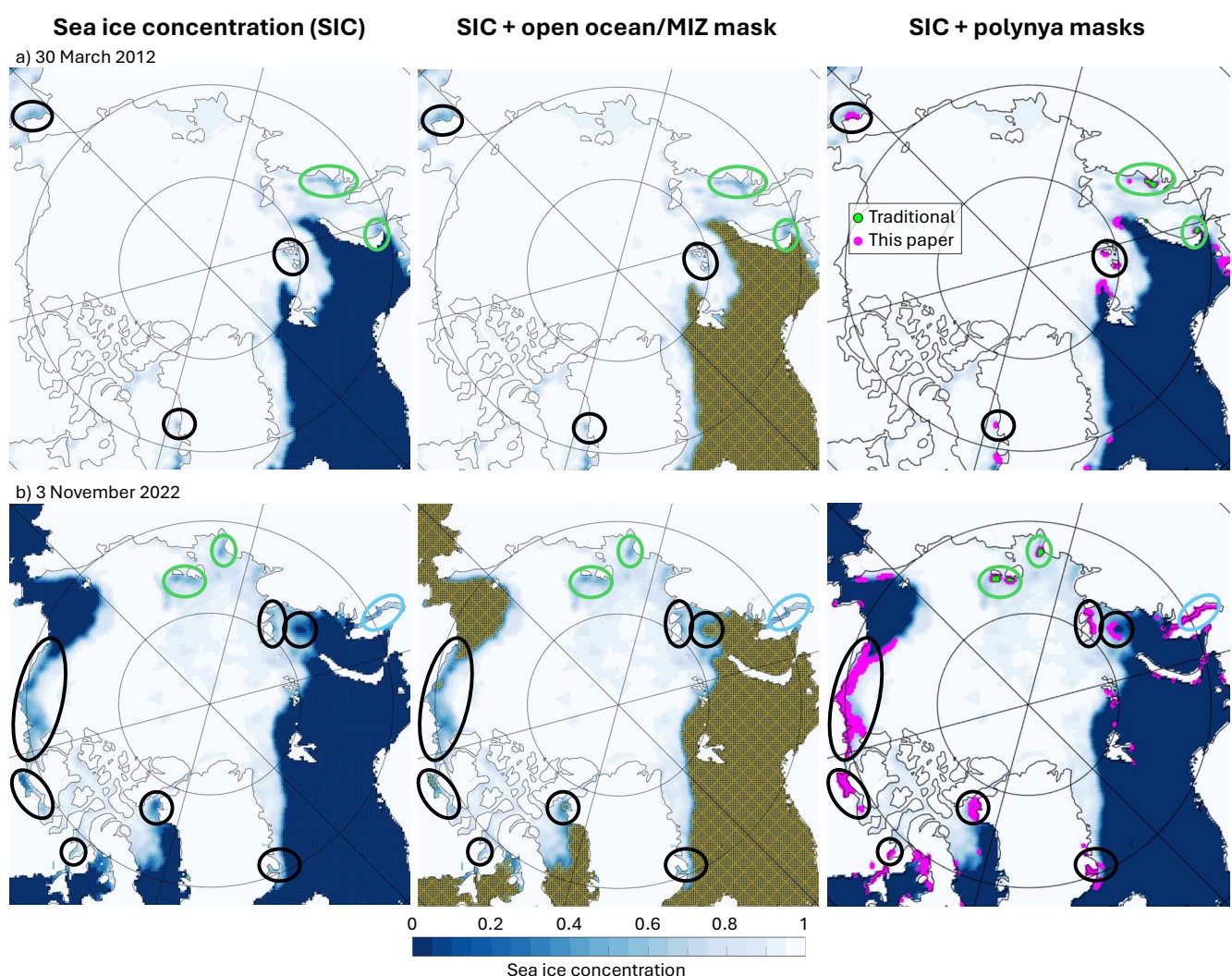

**Figure 5.** Two example results, on a) 30 March 2012 (top) and b) 3 November 2022 (bottom), of the original daily sea ice concentration data (left), with superimposed the resulting GMM-produced open ocean / MIZ mask (middle, orange asterisks) or the polynya masks (right, green and magenta). Green dots are the polynyas detected with a traditional threshold method, i.e. the labelled dataset we used for training, and magenta dots are the polynyas detected with our method. Green circles highlight areas where the two methods agree. Black circles highlight areas with false positives that we argue are correct, while the blue circle on b) is a special case, see text.

## 3.2 Validation: Variability in observed winter Arctic polynyas

Spatial and temporal variabilities and trends of winter Arctic polynyas are the topic of Wong et al. (2025)'s study. To further
validate our results, and in particular quantify the impact of our larger polynya areas, we here briefly look at some aspects of
the temporal variability in polynya activity, and compare our results to those of Wong et al. (2025). We here combine our true
positives and false positives, since we argued in the previous subsection that most of the false positives are actually polynya
pixels.

We first focus on the decadal variability (Fig. 6a-d). The frequency of polynya activity (Fig. 6a), i.e. how many years out of
the 10 in the decade has each pixel had at least once a polynya, highlights the same high-activity regions as previous studies (see
Introduction); the high activities east of Greenland and west/south of Svalbard are probably due to unfiltered MIZ. Comparing
each decade to the previous one reveals different variabilities in different regions. The Pacific side for example has an increased
activity in the 1990s (red on Fig. 6b), a paused change in the 2000s (pale colours on Fig. 6c), and an increase again in the 2010s
(red on Fig. 6d). On the Atlantic side in contrast, polynya activity is constantly moving out of the Barents Sea (blue colours),
either north into the Central Arctic or east into the Kara Sea (red colours), consistent with the all-season retreat of the sea ice
edge. Similarly east of Greenland experiences a decrease in the 1990s and 2000s and little change in the 2010s, because the sea
ice edge has already retreated. We suspect that this is the main reason for our reduced number of false positives in these regions
with time (Fig. 4): no polynya can be detected if there is no surrounding pack ice. Two other regions with known polynya
activity are west of Greenland and the Siberian Arctic, and they also have different rates of increased activity depending on the
decade (Fig. 6a-d). These results are in agreement with Wong et al. (2025), albeit with a different region definition: although
the overall pan-Arctic trend is towards an increased polynya area, the different regions increase at different rhythms because of
their respective forcing mechanisms (notably changes in the wind and air temperature). Explaining those would be beyond the
scope of this paper.

We also compare Wong et al. (2025)'s pan-Arctic trend to those obtained with our method (Fig. 6e and f). As in Wong et al.
(2025), we distinguish each year's "total" and "cumulative" polynya areas. The total area is the area of the Arctic that has had
at least one day with a polynya over the entire winter; the cumulative area is the sum of the daily pan-Arctic such areas, and
is therefore larger. As expected from our large number of false positives, i.e. pixels that were not detected as polynya in the
labelled training set created by Wong et al. (2025), the areas produced with our method (magenta lines) are larger than theirs
(black), about consistently three times as much. The trends are similar, although ours also reflects our increasing trends in false
positives and are therefore slightly larger: for the total area, 23 000 $\text{km}^2$ per year with our method, 17 000 with theirs over the
entire period; for the cumulative area, 106 000 $\text{km}^2$ per year with our method, 97 000 with theirs for years after the large jump
of 1988, synchronous in both series (198 000 and 120 000 respectively for the entire series). We suspect that the reason for this
jump is the change of program that year, from SMMR Nimbus-7 to SSM/I DMSP, but this is beyond the scope of this paper.
The series are also strongly and significantly correlated: 0.90 for the the total area, 0.89 for the cumulative area over the entire
period.

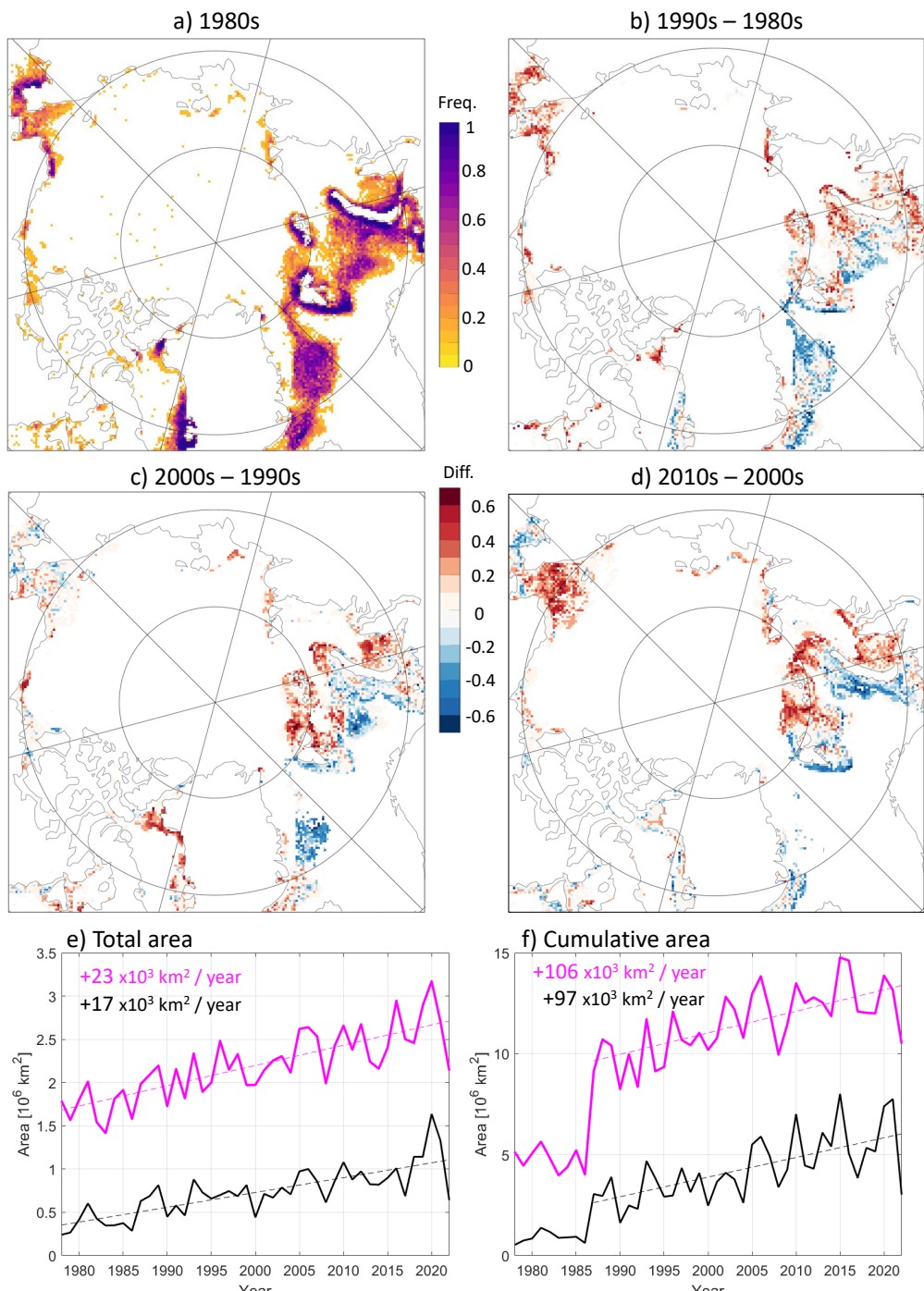

**Figure 6.** Variability exhibited by our results: Combining our true and false positives, for each pixel a) frequency in polynya opening in the 1980s, i.e. out of the 10 winters of that decade, how many had a polynya (only >0 shown); b-d) difference in polynya opening between two consecutive decades; e) total and f) cumulative polynya area of each winter in our results (magenta) and in Wong et al. (2025) (black) for the entire Arctic, along with the corresponding linear trends. See text for definition of each area.

In summary, as found in the previous subsection, our method detects larger polynya areas than the fixed 50% threshold method. This larger area does not significantly affect the geographical patterns of polynya activity and only slightly increases the temporal trends. The correlation between the methods is very strong.

## 3.3 Application: Detection of winter Arctic polynyas in the global coupled model MIROC6

The ultimate objective of this exercise was for us to develop a method for automatically and rapidly detecting Arctic polynyas in modelled sea ice concentration output. We therefore test our scripts on one global coupled model which we know has winter Arctic polynyas: MIROC6. We do not have any labelled data for it, so the assessment is based on visual inspection and comparison to fixed-threshold methods. We visually inspected every single monthly result, and a random quarter of the daily ones. As exemplified on Fig. 7 and Appendix Fig. A2, our model performs well, and equally well on the two temporal resolutions. As with the observations, the occasional false positive pixel still exists in the open ocean / MIZ filtering, but polynyas big and small are successfully detected. The fact that it works so well even on monthly data is surprising, given that both the brief and extensive studies of Heuzé et al. (2023) and Mohrmann et al. (2021), respectively, concluded that polynya detection on monthly modelled data is not recommended.

If anything, our visual inspection reveals that our results (magenta dots on Fig. 7) are somewhat conservative; we would have considered flagging as polynya the regions on the Siberian shelf with reduced SIC of around 70%. We therefore now focus on the Siberian shelf, more specifically the known "real world" polynya regions of the Laptev and Kara seas (see locations on Fig. 1). We compare the polynya area we obtained with our method to that from the range of SIC thresholds used in the literature (Fig. 8, green lines). Special attention is given to the 50% threshold that was used in the labelled observations the model was trained on (plain bright green). In the Laptev Sea, our method returns larger areas than the highest threshold of 60% for 21 out of the 30 years; in the Kara Sea, only 3 times. Those are small differences of the order of 10 pixels, and only when the total area is small (of the order of $100\,000\,\mathrm{km}^2$ in the Laptev Sea, Fig. 8a). In both regions, for the larger events, our method returns an area between those of the 50 and 60% thresholds. The daily areas confirm this picture: most of the values sit close to the unit line, but for both regions, the higher values are lower with a fixed 50% threshold than with our method (top-right corners of Fig. 8b and d). Regardless of the threshold, region, or whether we consider the cumulative or total areas, the correlations are very high between the series (often exceeding 0.9, Appendix Table A2), proving that our method captures well the variability in polynya activity.

Admittedly, one source of error of our method is that to directly apply our trained model, we had to interpolate the modelled data onto the NSIDC polar stereographic grid. We clearly introduced some interpolation artefacts (see the top left corner of all the panels, Figs. 7 and A2). One alternative could be to use only the unsupervised GMM to detect the open ocean / MIZ on the model's native grid, instead of the tedious traditional flood-fill, and then detect polynyas using a standard threshold method. This could be particularly advantageous if working with many models, since they have different sea ice concentration biases (Notz and Community, 2020). One inconvenience is the computing time: it takes less than a minute to apply the already-trained Unet to 30 years of daily data, but more than 15h on our supercomputer to obtain the classes from the GMM. The fastest approach of them all would then be to do the opposite and use only our trained Unet, and minimise the false positives

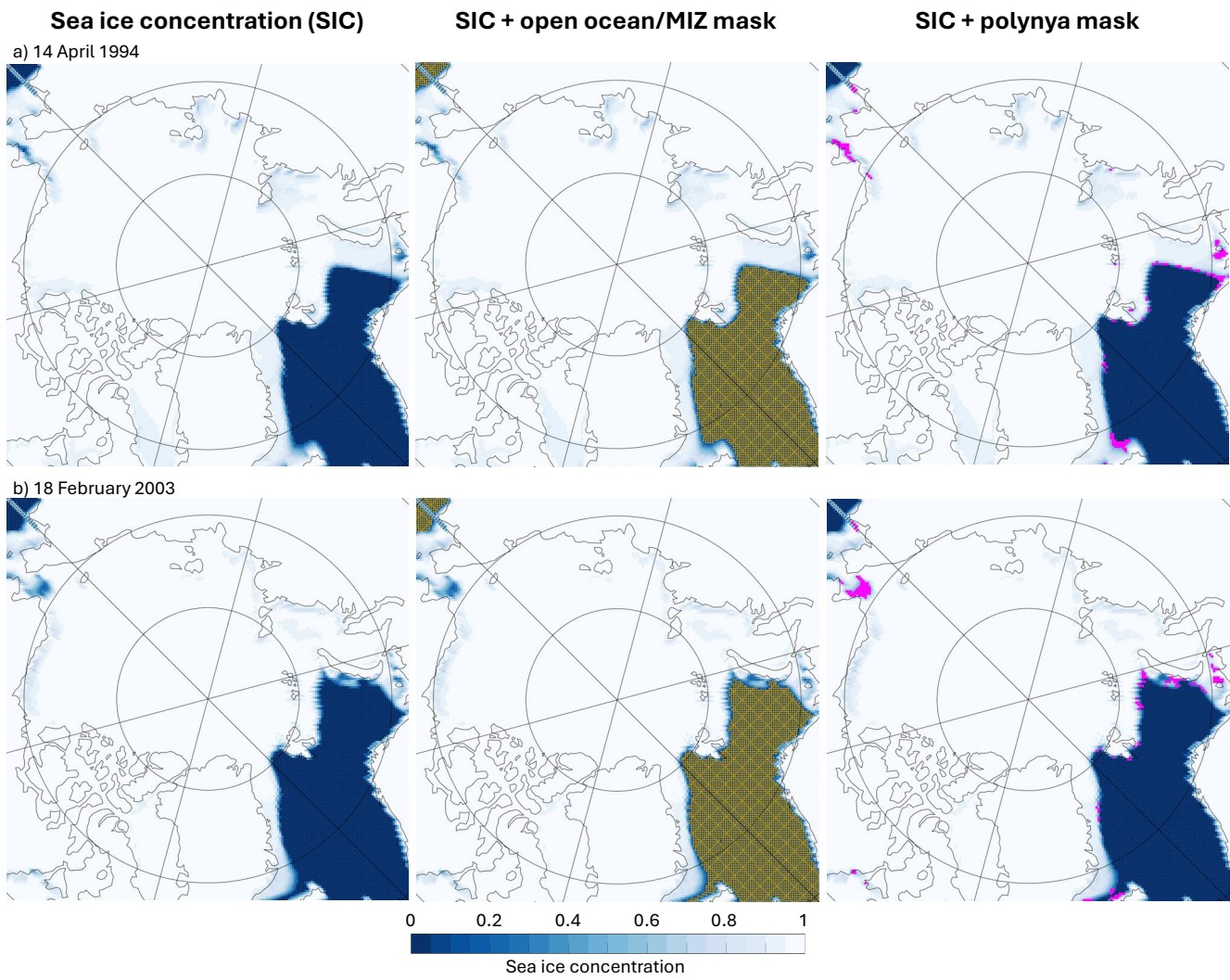

**Figure 7.** Two example results, on a) 14 April 1994 (top) and b) 18 February 2003 (bottom), of the model MIROC6's daily sea ice concentration data (left), with superimposed the resulting GMM-produced open ocean / MIZ mask (middle, orange asterisks) or the polynya mask (right, magenta).

by analysing pre-determined sub-regions in the pack ice, away from the MIZ, as done on observations by e.g. Tamura and Ohshima (2011) or Preußer et al. (2016).

We acknowledge that MIROC6 has one of the highest resolutions of all models that participated in the Climate Model Intercomparison Project phase 6 (CMIP6, Eyring et al., 2016), and has been identified in the all-model assessments of monthly and daily Arctic sea ice concentration of Athanase et al. (2025) and Heuzé and Jahn (2024), respectively, as among the most accurate CMIP6 models. Our results should be taken as a proof of concept: had our method not worked on MIROC6, it would

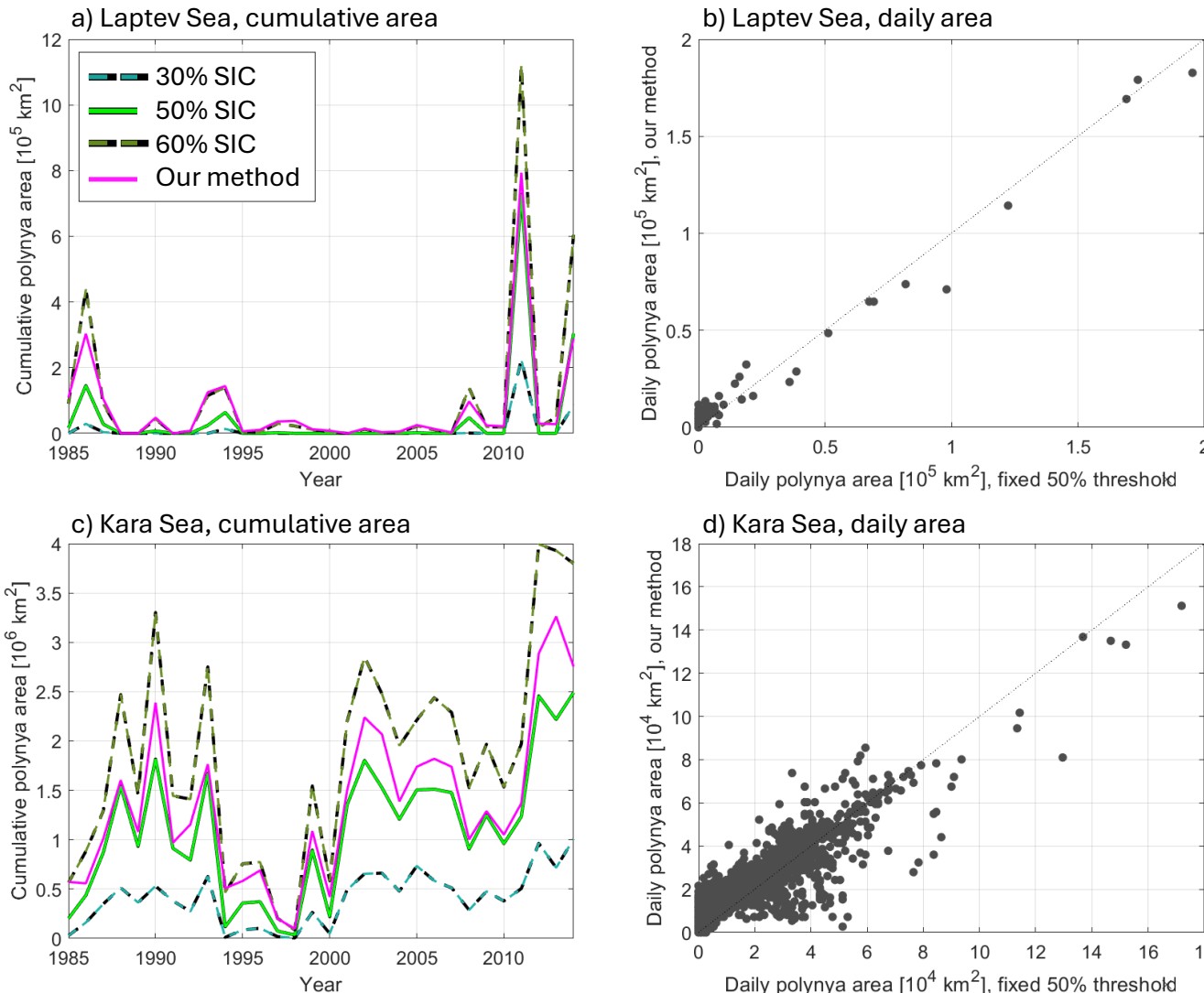

**Figure 8.** Polynyas in the Laptev (top) and Kara seas (bottom) in MIROC6 daily data. Left: Comparison of the cumulative winter polynya area retrieved by our method (magenta) and several commonly used fixed sea ice concentration thresholds (green). Right: Comparison of all the daily polynya areas between our method and the 50% threshold it was trained on; black line is the x=y unit line.

not have worked on the other CMIP6 models, but one will probably need to exercise caution if working with less accurate models. Daily sea ice thickness was not widely available in CMIP6; this is another reason why we developed our method on concentration instead. As with the observations, if daily thickness becomes routinely available for CMIP7 or if one wants to work with CMIP6 monthly thickness, we argue that our method can be used with minimum adjustment, provided one finds the
suitable corresponding thresholds.

## 4 Conclusions

We successfully trained a hybrid supervised-unsupervised machine learning model to automatically detect polynyas in the Arctic from daily satellite images of sea ice concentration, despite polynyas occupying barely 0.1% of the image. The supervised network is a Unet autoencoder, trained on the labelled polynya masks of Wong et al. (2025). It returns many false positives in the marginal ice zone, so we filter its results using an unsupervised Gaussian Mixture Model classifier that detects the open ocean / MIZ on each image. By combining the labelled sea ice concentration and thickness masks over the period where they are both available (since 2012), we reduce their individual noise and show that we have virtually no false negatives, i.e. all polynyas are detected. Our large number of false positives is to a small extent caused by the occasional complex unfiltered pattern in the MIZ, but is primarily due to the standard definition of a polynya. Our model detects areas with reduced sea ice cover surrounded by more compact ice, which we argue meet the criteria of a polynya, whereas traditional methods use fixed concentration thresholds for detection (e.g. Smedsrud et al., 2006; Tamura and Ohshima, 2011; Campbell et al., 2019). These not-that-false positives significantly increase with time (+277 per year) and are significantly anticorrelated to the Arctic sea ice area and extent (R = -0.76 and -0.71 for daily values), suggesting that as climate change continues impacting the sea ice cover (Stroeve and Notz, 2018), traditional definitions become less adapted, while the more flexible machine learning approaches can still successfully detect shapes and gradients in an icescape with shifted values.

As proof-of-concept, we applied our method to detect polynyas in one CMIP6 global coupled model, MIROC6. It indeed works, successfully removing the MIZ and detecting polynyas large and small on both daily and monthly modelled sea ice concentration. In the Laptev and Kara seas where we conducted more in-depth analyses, our method returned areas comparable to those with fixed thresholds between 50 to 60%, but the offset between the values is not constant - our method is more flexible. The correlation between the time series is very high regardless of the threshold used - our method preserves the variability, a result we also found when comparing to the observation results of Wong et al. (2025). This model has a high resolution, comparable to that of the observational product (Tatebe et al., 2019), and a somewhat realistic sea ice cover (Heuzé and Jahn, 2024); further studies will have to determine whether adjustments are necessary for coarser and/or more biased models. Similarly, we limited this study to sea ice concentration as observational sea ice thickness datasets (Paţilea et al., 2019) are not long enough yet to deal with such high-class-imbalance classification exercise, and to winter only since the labelled data covered only November to April each year (Wong et al., 2025). Nonetheless, this study paves the way for automatic detection at a somewhat low computing cost of Arctic polynyas, small features with such a large role for the Earth System (e.g. Smith Jr and Barber, 2007; Else et al., 2013; Moore et al., 2023).

*Code and data availability.* The scripts used in this manuscript and the best model we generated are available on Github at https://github.com/cheuze/Polynya_CNN2D_GMM; they will be licensed on Zenodo closer to publication. The observed sea ice concentration data from the National Snow and Ice Data Centre (dataset DOI 10.5067/MPYG15WAA4WX, DiGirolamo et al., 2022) are freely available at https://nsidc.org/data/nsidc-0051/versions/2, last accessed on 5 May 2025. The polynya masks produced by Wong et al. (2025) are in the process of becoming publicly available; in the meantime, they can be accessed at https://www.dropbox.com/scl/fi/fr0barqpe519e1nu6kl3q/

daily_location.nc?rlkey=zh1ganta0noexwvvvch4sh3zr&dl=1 and https://www.dropbox.com/scl/fi/9a91cbsgrqj29o5ckb5zq/daily_location_

SMOS.nc?rlkey=bdvlvay1k0rkql973u731ku0z&dl=1. The modelled sea ice concentration data from MIROC6 (dataset DOI 10.22033/ES-GF/CMIP6.5603, Tatebe and Watanabe (2018), last accessed 5 May 2025) are freely available on any of the Earth System Grid Federation portals; we used the French one: https://esgf-node.ipsl.upmc.fr/search/cmip6-ipsl/, last accessed on 5 May 2025. The regional mask from the National Snow and Ice Data Centre (dataset DOI 10.5067/CYW3O8ZUNIWC, subset "North, Polar Stereographic PS, 25 km", Meier and Stewart, 2023) is freely available at https://nsidc.org/data/nsidc-0780/versions/1, last accessed on 5 August 2025.

**Appendix A:  Appendix figures and tables**

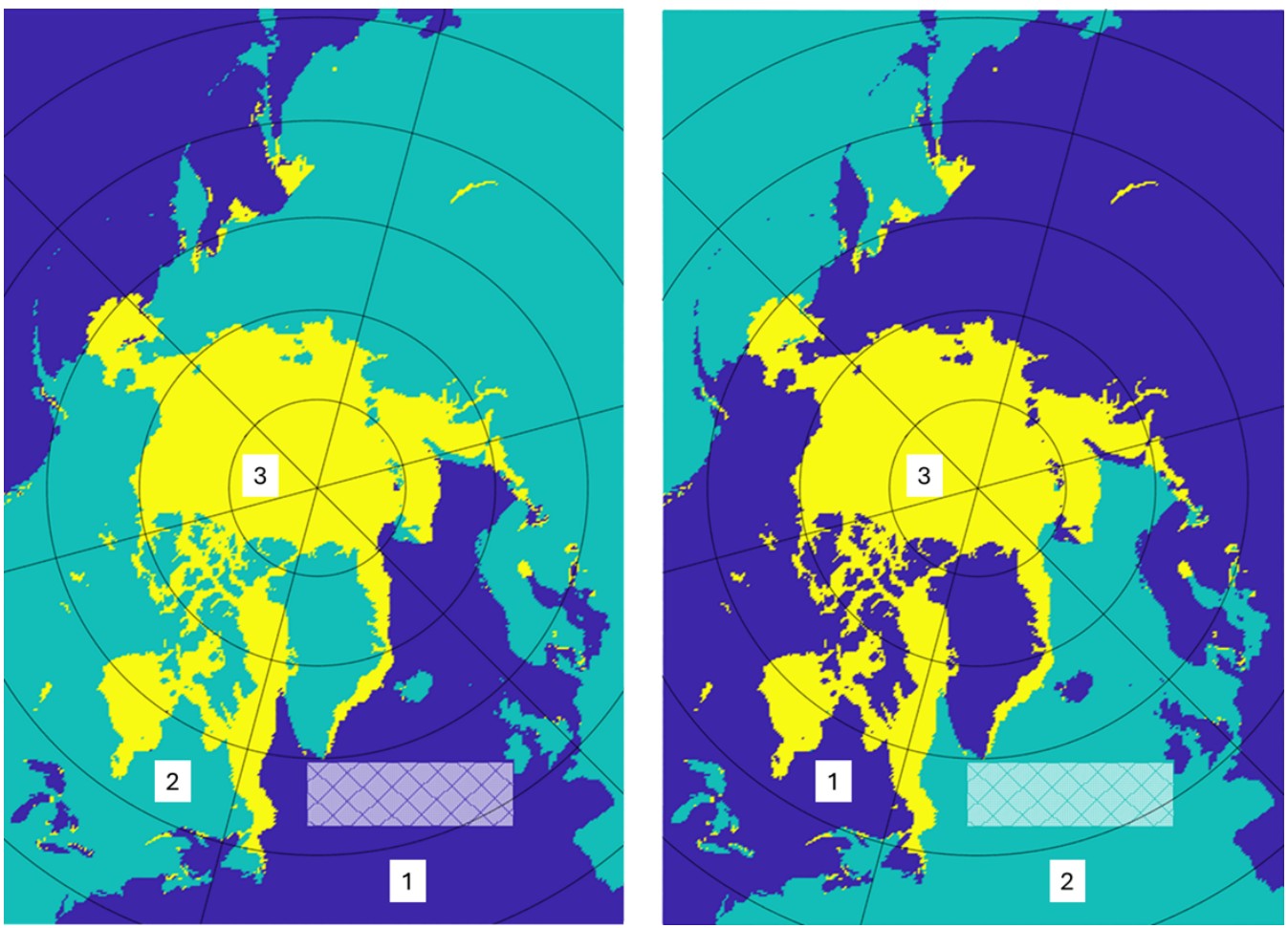

**Figure A1.** Two runs of the GMM unsupervised classification on the same random timestep (29 April 1984), illustrating that classes at the same location can randomly be assigned different numbers: Here classes 1 (dark blue) and 2 (green) are swapped. The white band over the North Atlantic is that used to detect the open ocean / MIZ class number.

|  | Nov-Dec | Jan-Feb | Mar-Apr |
|---|---|---|---|
| 1. Central Arctic Ocean | 0.15 | 0.25 | 0.13 |
| 2. Beaufort Sea | 0.35 | - | - |
| 3. Chukchi Sea | 0.19 | 0.08 | 0.17 |
| 4. East Siberian Sea | 0.15 | - | - |
| 5. Laptev Sea | 0.05 | 0.02 | 0.04 |
| 6. Kara Sea | 0.40 | 0.30 | 0.19 |
| 7. Barents Sea | -0.28 | -0.20 | - |
| 8. East Greenland Sea | -0.17 | -0.72 | -0.52 |
| 9. Baffin Bay | 0.18 | 0.18 | 0.25 |

**Table A1.** Trend in false positives, in pixel per year, for the nine regions and three time periods shown on Fig. 4. Only trends significant at 95% are shown. See also region definition on Fig. 1.

| SIC threshold | Laptev Sea | | Kara Sea | |
|---|---|---|---|---|
| | cumul. | daily | cumul. | daily |
| 30 | 0.95 | 0.95 | 0.93 | 0.86 |
| 50 | 0.97 | 0.98 | 0.97 | 0.92 |
| 60 | 0.98 | 0.96 | 0.99 | 0.95 |

**Table A2.** Correlation coefficient R between the cumulative (cumul.) or daily polynya areas in MIROC6 obtained using a fixed sea ice concentration threshold and that using our method, for the Laptev and Kara seas.

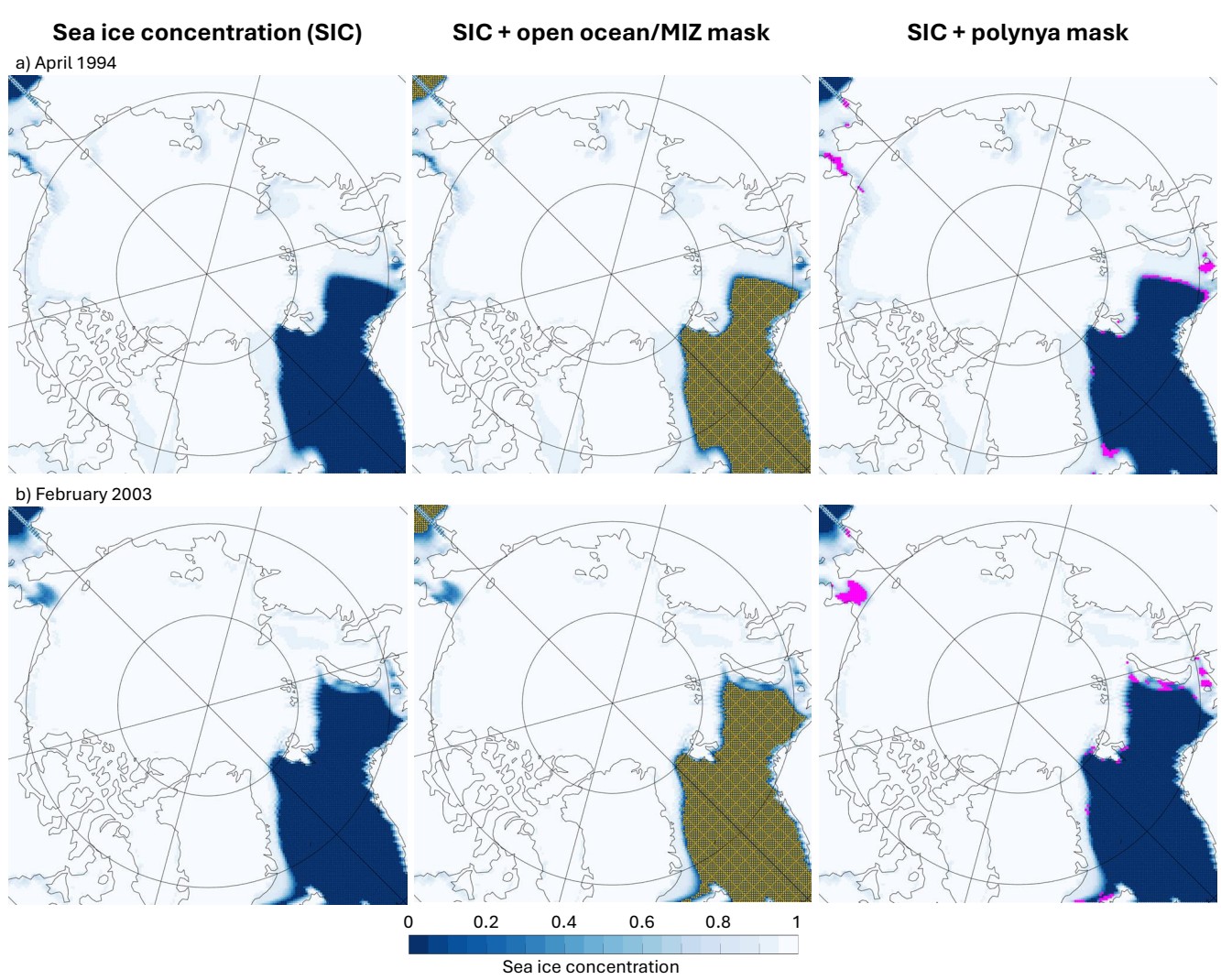

**Figure A2.** Same as Fig. 7 but using the model's monthly mean sea ice concentration.

*Author contributions.* CH designed and conducted most of the study. CHMW obtained the data and produced the labelled, training dataset. CH wrote the original draft. Both authors revised the manuscript.

*Competing interests.* The authors declare that they have no conflict of interest.

*Acknowledgements.* This work was funded by the Swedish National Space Agency grant no. 2022-00149 awarded to CH. The model training was enabled by resources provided by the National Academic Infrastructure for Supercomputing in Sweden (NAISS), partially funded by the Swedish Research Council through grant agreement no. 2022-06725. We acknowledge the World Climate Research Programme, which, through its Working Group on Coupled Modelling, coordinated and promoted CMIP6. We thank the climate modeling groups for producing and making available their model output, the Earth System Grid Federation (ESGF) for archiving the data and providing access, and the multiple funding agencies that support CMIP6 and ESGF. We also thank the two anonymous reviewers for their time commenting on this manuscript.

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
