# Peer review of "Automatic detection of Arctic polynyas using hybrid supervised-unsupervised deep learning"

_EGUsphere, 2025_

## Referee Comment (RC1)

**Review on "Automatic detection of Arctic polynyas using hybrid supervised-unsupervised deep learning" by Céline Heuzé and Carmen Hau Man Wong** // submitted to The Cryosphere // July 2025

The authors present a study that uses a labelled dataset (daily polynya masks) to train a machine learning (ML) model to detect Arctic polynyas (areas with open water and/or thin sea ice) in sea ice concentration (SIC) maps (based on PMW remote sensing data). Presented results indicate that the model was able to identify polynyas with minimal false negatives, and most "false positives" were claimed by the authors to be correct detections of patterns with reduced SIC. The model was then also applied to the output of a particular climate model of the CMIP6 range, but the authors highlighted that there are several challenges attached to this kind of endeavour, i.e. identifying polynyas in more or less coarse resolution output fields with a (so far) varying level of detail in terms of sea ice information. Nevertheless, the presented approach and its first results make it a promising and seemingly adaptable tool for future studies that aim to continue the monitoring of Arctic polynyas in a rapidly changing environment.

The paper is very well written, reads fluently and the quality of the figures is overall good, with a few suggestions for further improvements given below as part of my "Specific comments". Otherwise, please also find my more general remarks and comments below. Overall, I consider the study to be highly interesting and hence very well publishable, but only after some more or less major revisions, additions and clarifications are addressed by the authors. These identified issues mainly revolve around some parts of the manuscript that would benefit from more care/detail in explaining the taken approach with its advantages/limitations, but also include the overly short discussions and comparisons on the thereby identified (interannual) polynya occurrences in the Arctic.

**General comments**

(1) Regarding the anti-correlation of false-positive polynya pixels (which the authors mostly consider as true) with SIE/SIA: the authors could comment on the circumstance, that the increasing number of polynya pixels (Fig.4) is likely related to another rather simple reason – the delayed fall freeze-up in many regions at or in proximity of the MIZ. In the present manuscript, I often had the feeling that it is not really separated between the regular freeze-up in late fall (~NOV/DEC), and "real" wind-driven polynya events in an otherwise more rigid ice regime. While this is certainly an issue for most studies on polynya detection in the Arctic during winter, I would at least appreciate a few additional sentences in the manuscript that properly addresses these methodical challenges.

(2) A bit related to my comment above: This issue also translates to the presented application to detect wintertime polynyas in MIROC6. There, the authors consider their model to perform well and note that "polynyas big and small are successfully detected". However, when taking a closer look at the presented SIC maps in Fig. 6 and 7, my impression is that most marked polynya areas are more likely part of remaining regions with MIZ characteristics (despite the GMM approach), while other areas of decreased SIC at well-known sites of polynya formation (such as the Laptev and Kara Seas, northern Baffin Bay, Beaufort Sea) are missed. I think it is great to see that the taken approach is somewhat flexible in terms of input data, but as the authors rightly acknowledge themselves – in this form it is purely a proof-of-concept that certainly requires some subsequent fine-tuning.

(3) In terms of presented results, I'm missing some sense of the long-term performance on annual / interannual timescales. A straight forward addition would be for instance maps of polynya occurrences (per winter or decadal averages) that would then help to identify if/how the here detected polynya sites around the Arctic compare to previous assessments in published literature. I noticed that the preprint by Wong et al. concentrates on that aspect (and more), so I guess such a comparison and related discussion should be rather easy to include here as well. It would also increase the impact of your study, beyond the current technical focus.

**Specific comments**

*Abstract*

L.1: "…areas with no- or thin-ice" → more commonly "open water and thin-ice"

L.11: "that the rigid traditional methods with fixed thresholds cannot identify" → not that I'm asking for a long explanation in the Abstract (please don't), but I'm asking myself – why exactly?

*Introduction*

L.15: "Polynyas, small openings in the pack ice, …" → in what sense small? Maybe worth to add some sort of reference for typical polynya sizes

L.17: Regarding the references for SIP in the Arctic – both of these cited papers had several follow-up studies over the years that introduced some changes/improvements to certain methodical aspects. Any particular reason why you picked these two?

L.19: "…and even impacts the large-scale atmospheric circulation" → not only large scale, but also regional effects on atmospheric boundary layer dynamics. Likely implied by "strong heat loss to the atmosphere" a few lines above, but not explicitly stated/referenced (e.g., Marcq and Weiss, 2012; Lüpkes et al., 2008).

L.24: I wouldn't use the terms "easy" and "way more tedious" here – that sounds like downplaying the referenced study. Maybe try to find a more "diplomatic" phrasing.

L.34: Why shouldn't they remain valid, and in what sense? Please explain your thoughts here.

L.38: "6 months" → six months

L.45: You could add a note here that this data set will be explained in a bit more detail in Section 2.1

L47: "Following Liu et al. (2025)'s work…

*Data & Model architectures*

L.56: In my opinion – this single sentence is not really necessary here as it just partly repeats the end of section one.

L.60-62: Why did you use this (now rather legacy) data set and not (for instance) the OSI-SAF equivalent? And: did you also try out alternative PMW data sets with higher spatial resolutions at some point? It would be interesting to see how your approach fares in regions where the grid resolution makes a real difference.

L.63: "no threshold for the area" → but I assume any pixel with a SIC > 0%, right? So not exactly *no* threshold.

L.67: "Briefly" → "In brief,"

L.84: "all NaNs" – I assume these contain the "pole hole" (basically filled up with 100% SIC as it seems?), land and missing pixels – something else? Details not given here.

L.86: "polynyas are rare events" → not true for every polynya, for instance the NOW polynya. If it's more or less open at more than 50% of the time during winter, I wouldn't call it rare. "Used to be rare" could be true however.

L.88: How do you define "scenes" here? 192 x 192 pixel grids or smaller subsets?

L.118: "the number of false negatives was not significantly affected" → any idea why?

L.136: "2 to 10 images" → i.e., 2 to 10 *daily* images, correct?

Figure 3: To be honest, I'm not sure if that large direct comparison of different class numbers is really adding much to the overall story. It could be more interesting to get an insight into seasonal differences in the GMM performance, and/or comparisons to other independent data sets.

*Results and Discussion*

Table 1 (caption): "269 million pixels in total" – what is total? All pixels or just those containing ocean?

L.167/171: "Fig.5" → Fig.4?

L.169: "the less sea ice, the more false positives" – any explanation for that (relates to my initial general comments)?

Figure 4: As they were mentioned in the text → one could add the corresponding regression lines and related statistics

L.187-189: "polynyas become harder to predict with traditional methods" → I can't defer that from Fig. 4 and Fig.5 alone, as you did not differentiate / illustrate the long-term evolution of the "traditional" threshold methods. And do you really mean "predict", and not rather "detect"?

L.192: "sea ice thickness data" → Please indicate which SIT you mean here, as other derived SIT estimates were obviously also utilized in earlier studies, incl. those prior to 2010

L.202: MIROC6 → reference missing;

L.202: "we don't have any truthing data for it" → That admittedly sounds a bit handwavy, so can you at least try to list some model characteristics that could be behind this circumstance? In know that grid resolution is one prominent candidate here (as discussed towards the end of section 3.2), but maybe there's more to it.

Figures 5, 6 and 7: Lat/lon values not indicated, should be added in at least one panel per figure. Fig. 6 and 7 are missing a clear depiction for the magenta colour, as in Fig. 5.

*Conclusions*

L.231: "It returns many false positives in the marginal ice zone, so we filter its results using an unsupervised Gaussian Mixture Model classifier that detects the open ocean / MIZ on each image." → This effect could have been illustrated somewhere in Section 2.

L.234: "all polynyas are detected" → how can you be sure about that?

L.244: "that it indeed works" → "that it indeed works *to some extent*."

---

## Referee Comment (RC2)

**Review - Automatic detection of Arctic polynyas using hybrid supervised-unsupervised deep learning**

This is an interesting proof-of-concept study building on a recently published labelled dataset to detect Arctic polynyas using deep learning. The concept is novel and the manuscript is concise and reads smoothly. The limitations of the experiments are clearly outlined. The method, despite its limitations, demonstrates the potential to be further adapted to become a full-fledged machine learning based automatic polynya detector for future Arctic climate studies. Therefore, I consider the manuscript publishable with some revisions, especially regarding the theoretical grounding of the proposed shift in the definition of polynyas. Strengthening the assessment of the method's performance when applied on climate model output is also needed. I attach my general and more specific comments as follows.

**Major comments**

1. I'm missing some deeper and more systematic discussions on the definition of polynyas in relation to all the analyses done, since through the manuscript this seems to have shifted somewhat, but it is fundamental to the experiment design. An example where such discussions could help to address: to what extent is the argument 'areas where SIC strongly decreases compared to the surrounding ice can be defined as polynyas' valid, so that the 'not-that-false positives' could be deemed as 'true positives?'

   One would intuitively think that the term is geographically based and would not shift with time like the authors argue for. I appreciate that perhaps the authors are pointing towards shifting the definition to a comparative one rather than an absolute one, but this should be more clearly explained. As of now, this argument seems to be 'data-driven', i.e., a change in definition is needed as the model detects 'reduced sea ice cover surrounded by more compact ice.' However, the readers perhaps would like to first know why such a shift in definition is theoretically valid and important within the scope of sea ice/climate science.

   Some more discussions on defining polynyas in the Introduction could help set the stage for these arguments. Then, strengthening these arguments in the Results and Discussion will also be desirable.

2. Lines 200-206 & Fig. 6-7: I would be hesitant to agree with the polynyas detected along the MIZ in the Atlantic Sector, and would recommend more discussions on if these are false positives, and if similar polynya detections are common in the time series. Also, I feel Figs. 6-7 do not convey enough information to do justice to the authors' analyses. For example, the authors could present a monthly time series of a cropped-in region of interest (e.g. an area of known polynya formation) in a given year in tightly-packed sub-figures, or a time series for a given month/date for all the years.

3. It would be good to know how the method transfers to the AMSR-E based 12.5km SIC product by NSIDC. If re-running on some of these data is not within the scope of the paper, some mentioning of this in the Discussions would be helpful for the readers to understand the potential of your methods combined with this SIC product with improved spatial resolution.

**Minor comments**

Line 4: 'from 0.1 to 2.5%' could be confusing. Perhaps 'from 10% to 2.5%.'

Line 16: 'not begin until' is a bit awkward. Consider rewording.

Line 40: "'zooms' on the image": I suppose you mean "'zooms in' on the image"

Line 131: what are the considerations going into keeping a 'land' class rather than directly applying a land mask and classifying pack ice vs open ocean? Could using a land mask help solve the issues encountered in Lines 132-133 and in Lines 147-149?

Line 144 & in Fig. 3 caption: randomly affected → randomly assigned?

Line 157: consider moving '(Table 1, first column)' to after 'The performances' to avoid confusion.

Fig. 4 caption: '2023 stops in April' → '2023 data stops in April.'  'higher sea ice values' → 'higher values in sea ice area and extent'

Line 180: 'a polynya' → 'polynyas'

Line 180: 'strongly decreases compared to' could be confusing which leads the reader to think of the time dimension. Consider changing to e.g. 'is significantly less than'

Line 181: the sentence after 'is not perfect' is confusing to read. Consider rewording. Do you mean 'the MIZ mask filters out some polynya pixels whose surrounding ice also has relatively low SIC, while in some complex MIZ regions, the MIZ mask fails to filter out some false positives?'

Line 203: 'we do not have any truthing data for it' – if not truthing, perhaps some comparisons with observation-based polynya detections in Section 3.1?

Line 204: 'the occasional false positive pixel survives' → 'occasional false positive pixels survive'

Line 214: 'The one inconvenient' → 'One inconvenience'

Line 234: 'no false negative' → 'no false negatives'

Throughout the manuscript:

'Timeseries' should be changed to 'time series'

After the first usage of 'SIC', avoid using 'sea ice concentration' to be consistent.

---

## Author Response (AR1)

Response to reviewers

We thank the reviewers for their comments. We have addressed them all, as detailed in this document. The reviewers' contribution has also been added to the acknowledgement section.

Reviewer 1

General comments

(1) Regarding the anti-correlation of false-positive polynya pixels (which the authors mostly consider as true) with SIE/SIA: the authors could comment on the circumstance, that the increasing number of polynya pixels (Fig.4) is likely related to another rather simple reason – the delayed fall freeze-up in many regions at or in proximity of the MIZ. In the present manuscript, I often had the feeling that it is not really separated between the regular freeze-up in late fall (~NOV/DEC), and "real" wind-driven polynya events in an otherwise more rigid ice regime. While this is certainly an issue for most studies on polynya detection in the Arctic during winter, I would at least appreciate a few additional sentences in the manuscript that properly addresses these methodical challenges.

We thank the reviewer for raising this valid concern. We have added a new figure (Fig. 4) and its corresponding appendix table (Table A1) to address this point. For the nine commonly defined Arctic regions (the marginal seas + the Central Arctic Ocean), for each year, we show the number of false positives during the freeze-up period Nov-Dec, the consolidated period Jan-Feb, and the potential early-melt period Mar-Apr. As explained in the core of the text accompanying this new figure, in short, no the increasing number of polynya pixels is not just an increase in mis-detection of the MIZ. Most of this increase takes place in Mar-Apr, consistent with Wong et al. (2025)'s finding of an increased polynya activity season, or in Jan-Feb when the sea ice is consolidated. We even find a strong decrease in the East Greenland and Barents seas, which along with the extra analysis we performed in response to your major comment (3), suggests that our method becomes better at filtering out the MIZ as sea ice retreats.

(2) A bit related to my comment above: This issue also translates to the presented application to detect wintertime polynyas in MIROC6. There, the authors consider their model to perform well and note that "polynyas big and small are successfully detected". However, when taking a closer look at the presented SIC maps in Fig. 6 and 7, my impression is that most marked polynya areas are more likely part of remaining regions with MIZ characteristics (despite the GMM approach), while other areas of decreased SIC at well-known sites of polynya formation (such as the Laptev and Kara Seas, northern Baffin Bay, Beaufort Sea) are missed. I think it is great to see that the taken approach is somewhat flexible in terms of input data, but as the authors rightly acknowledge themselves – in this form it is purely a proof-of-concept that certainly requires some subsequent fine-tuning.

Reviewer 2 made a similar comment. This team is presently conducting an in-depth analysis of CMIP6 models' representation of Arctic polynyas, and doing so here would be beyond the scope of this paper. One main finding, which is not unique to Arctic polynyas, is that not detecting them at known "real world" sites is often not a detection issue but a model bias.
Nonetheless, Figs 6 and 7 do exhibit reduced sea ice cover and the occasional CNN-detected pixel at these sites, so in response to your and Reviewer 2's comment we have now expanded this subsection and added a new figure, Fig. 8, and Appendix table A2, dedicated to polynya activity in MIROC6 in the Laptev and Kara seas. We there compare the performances of our method to fixed-threshold detections to complement the visual assessment. We find a strong correlation often exceeding 0.9

regardless of the diagnostic, region, or SIC threshold we compare it to, showing that our method captures very well the variability in modelled polynya activity in the region. Our polynya areas often fall between those of the 50 and 60% thresholds, but the offset is not constant, indicating again that our method is comparatively flexible.

(3) In terms of presented results, I'm missing some sense of the long-term performance on annual / interannual timescales. A straight forward addition would be for instance maps of polynya occurrences (per winter or decadal averages) that would then help to identify if/how the here detected polynya sites around the Arctic compare to previous assessments in published literature. I noticed that the preprint by Wong et al. concentrates on that aspect (and more), so I guess such a comparison and related discussion should be rather easy to include here as well. It would also increase the impact of your study, beyond the current technical focus.

These are indeed the topic of Wong et al.'s study, and we do not want to compromise their chances of publication by presenting another version of their results here. However, following your suggestion, we added a new subsection, subsection 3.2, entitled "Validation: Variability in observed winter Arctic polynyas". As its title suggest, we verify in this section how some of the temporal variabilities presented in Wong et al. are modified by our false positives. The main results are that the decadal variability is in strong agreement with that of Wong et al., with the different regions increasing their polynya activity at different time periods. Wong et al. explains that this is in response to the regions' respective forcings. Our pan-Arctic trends are slightly larger than theirs, as was to be expected from our large trend in "false" positives, but the correlation between their and our series is very strong.

Specific comments

Abstract

L.1: "…areas with no- or thin-ice" → more commonly "open water and thin-ice"

Rephrased

L.11: "that the rigid traditional methods with fixed thresholds cannot identify" → not that I'm asking for a long explanation in the Abstract (please don't), but I'm asking myself – why exactly?

This joins Reviewer 2's first major comment. In my opinion, this is because there is no strict definition of a polynya, but rather a series of processes whose effect, individual or combined, result in a reduction of sea ice within the pack ice. Deep learning is made for detecting patterns and relationships, and that is probably why it performs best.

Introduction

L.15: "Polynyas, small openings in the pack ice, …" → in what sense small? Maybe worth to add some sort of reference for typical polynya sizes

We added an upper limit, based on the typical size of the largest Arctic polynya, the North Water polynya.

L.17: Regarding the references for SIP in the Arctic – both of these cited papers had several follow-up studies over the years that introduced some changes/improvements to certain methodical aspects. Any particular reason why you picked these two?

The lead author likes these two papers. Since sea ice production is not the topic of this current manuscript and the cited literature is only for illustration purposes, the methodological changes are not crucial here.

L.19: "...and even impacts the large-scale atmospheric circulation" → not only large scale, but also regional effects on atmospheric boundary layer dynamics. Likely implied by "strong heat loss to the atmosphere" a few lines above, but not explicitly stated/referenced (e.g., Marcq and Weiss, 2012; Lüpkes et al., 2008).

We have clarified this sentence by changing to "regional and large-scale" instead of just "large-scale".

L.24: I wouldn't use the terms "easy" and "way more tedious" here – that sounds like downplaying the referenced study. Maybe try to find a more "diplomatic" phrasing.

The lead author of this present study also led the study that was being downplayed. We nonetheless rephrased.

L.34: Why shouldn't they remain valid, and in what sense? Please explain your thoughts here.

This part has been rephrased in response to the first major comment of Reviewer 2.

L.38: "6 months" → six months

Ok

L.45: You could add a note here that this data set will be explained in a bit more detail in Section 2.1

Done

L47: "Following Liu et al. (2025)'s work...

We do not understand this comment; we see no typo.

Data & Model architectures

L.56: In my opinion – this single sentence is not really necessary here as it just partly repeats the end of section one.

Ok, removed

L.60-62: Why did you use this (now rather legacy) data set and not (for instance) the OSI-SAF equivalent? And: did you also try out alternative PMW data sets with higher spatial resolutions at some point? It would be interesting to see how your approach fares in regions where the grid resolution makes a real difference.

For consistency, we used the same product as was used by Wong et al. to produce the labelled dataset. When they started their study in 2022, they could not foresee the result of the US presidential elections and that the NSIDC product would become legacy. The main difference between the two products is that NSIDC algorithms use fixed tie-points, whereas OSI-SAF uses dynamic ones. Since Wong et al. wanted to perform a long-term trend analysis, to minimise extra sources of uncertainty beyond the many choices of thresholds to use (in sea ice, in the region definition, in the seasons), they went for the NSIDC data. They also wanted to compare their results to older studies, most of which were based on the NSIDC data.

As for the second part of your comment, Wong et al. compares the NSIDC-based results to those produced with higher spatial resolution datasets. We do not, because high resolution means short time coverage and we ran out of computing time to test this now, but have added some text at the end of section 3.1 to explain how this could be done.

L.63: "no threshold for the area" → but I assume any pixel with a SIC > 0%, right? So not exactly no threshold.

Since pixels where SIC = 0 are going to add exactly 0 to the total, there is no need to set a threshold excluding them.

L.67: "Briefly" → "In brief,"

Ok

L.84: "all NaNs" – I assume these contain the "pole hole" (basically filled up with 100% SIC as it seems?), land and missing pixels – something else? Details not given here.

Details now given.

L.86: "polynyas are rare events" → not true for every polynya, for instance the NOW polynya. If it's more or less open at more than 50% of the time during winter, I wouldn't call it rare. "Used to be rare" could be true however.

We here used the machine learning terminology. The distinction is that the occurrence of interest is spatial, not temporal as the reviewer understood. Polynyas are rare events in the sense that on each scene (daily image), polynya pixels occupy a minuscule fraction of the image. We have rephrased to make this meaning clearer.

L.88: How do you define "scenes" here? 192 x 192 pixel grids or smaller subsets?

Size added

L.118: "the number of false negatives was not significantly affected" → any idea why?

We suspect that this is related to one of the first results we present in section 3.1: the false negatives are actually errors in the labelled datasets.

L.136: "2 to 10 images" → i.e., 2 to 10 daily images, correct?

"daily" added

Figure 3: To be honest, I'm not sure if that large direct comparison of different class numbers is really adding much to the overall story. It could be more interesting to get an insight into seasonal differences in the GMM performance, and/or comparisons to other independent data sets.

This figure was in the Methods section, and therefore was meant as a simple illustration of how the method works, also defining the region used for the class matching. Follower your comment, we decided to move this figure to the appendix.

Results and Discussion

Table 1 (caption): "269 million pixels in total" – what is total? All pixels or just those containing ocean?

All pixels – clarification added to the text

L.167/171: "Fig.5" → Fig.4?

Corrected when necessary; text clarified otherwise.

L.169: "the less sea ice, the more false positives" – any explanation for that (relates to my initial general comments)?

See response to the corresponding major comment.

Figure 4: As they were mentioned in the text → one could add the corresponding regression lines and related statistics

We originally did this but the figures were too cluttered, hence the mention in the text instead. We did include the regression lines and statistics to our new figure 6.

L.187-189: "polynyas become harder to predict with traditional methods" → I can't defer that from Fig. 4 and Fig.5 alone, as you did not differentiate / illustrate the long-term evolution of the "traditional" threshold methods. And do you really mean "predict", and not rather "detect"?

We indeed meant "detect" and have corrected. See the new analyses in response to your major comments for the time evolution and comparison with time of the different methods.

L.192: "sea ice thickness data" → Please indicate which SIT you mean here, as other derived SIT estimates were obviously also utilized in earlier studies, incl. those prior to 2010

In an earlier draft, this sentence read "good quality sea ice thickness data", which, to re-use your earlier words, was not very diplomatic. In response to your comment, we have rephrased as "adequate".

L.202: MIROC6 → reference missing;

The reference in the methods section where the model was described.

L.202: "we don't have any truthing data for it" → That admittedly sounds a bit handwavy, so can you at least try to list some model characteristics that could be behind this circumstance? In know that grid resolution is one prominent candidate here (as discussed towards the end of section 3.2), but maybe there's more to it.

This section has been rewritten in response to your second major comment and now includes quantified analyses.

Figures 5, 6 and 7: Lat/lon values not indicated, should be added in at least one panel per figure. Fig. 6 and 7 are missing a clear depiction for the magenta colour, as in Fig. 5.

The lat/lon values are on Figure 1 and not indicated again afterwards since the figures are already very busy. We will let the copy editors decide on the map style when the time comes. The colours for the figures that are now 7 and A2 are explained in the caption; figure 5 had a separate explanation because two methods were compared hence two colours are used to represent polynyas.

Conclusions

L.231: "It returns many false positives in the marginal ice zone, so we filter its results using an unsupervised Gaussian Mixture Model classifier that detects the open ocean / MIZ on each image." → This effect could have been illustrated somewhere in Section 2.

This is illustrated in the figure that in response your comment we have now moved to the appendix (Figure A1) and is quantified in Table 1.

L.234: "all polynyas are detected" → how can you be sure about that?

The whole sentence reads "we have virtually no false negatives, i.e. all polynyas are detected". We are sure that we detected all the polynyas because we have no false negative.

L.244: "that it indeed works" → "that it indeed works to some extent."

We have modified this sentence since we added new analyses related to this result in response to your major comment.

Reviewer 2

Major comments

1. I'm missing some deeper and more systematic discussions on the definition of polynyas in relation to all the analyses done, since through the manuscript this seems to have shifted somewhat, but it is fundamental to the experiment design. An example where such discussions could help to address: to what extent is the argument 'areas where SIC strongly decreases compared to the surrounding ice can be defined as polynyas' valid, so that the 'not-that-false positives' could be deemed as 'true positives?' One would intuitively think that the term is geographically based and would not shift with time like the authors argue for. I appreciate that perhaps the authors are pointing towards shifting the definition to a comparative one rather than an absolute one, but this should be more clearly explained. As of now, this argument seems to be 'data-driven', i.e., a change in definition is needed as the model detects 'reduced sea ice cover surrounded by more compact ice.' However, the readers perhaps would like to first know why such a shift in definition is theoretically valid and important within the scope of sea ice/climate science. Some more discussions on defining polynyas in the Introduction could help set the stage for these arguments. Then, strengthening these arguments in the Results and Discussion will also be desirable.

We carefully re-read the absolute reference on this topic, the 2007 book "Polynyas: Windows to the world" by Smith Jr and Barber, which confirmed our suspicion: There is no strict definition of what a polynya is. Instead, there is a wide range of processes that cause polynyas (shifting wind speeds and directions, air temperature, atmospheric rivers, ocean temperature and currents, vertical mixing in the ocean, ice internal stress and rheology) and an even wider range of processes that are impacted by polynyas (e.g. heat, moisture and gas fluxes to the atmosphere or ocean, ecosystem processes throughout the entire marine food web, vertical mixing in the ocean or atmosphere, water mass transformation). Without wanting to sound philosophical, one usually defines a polynya based on these processes and on which question the researcher wants to address at that point. This is most likely the reason behind the wide range of sea ice concentration thresholds used in the literature: even just a small decrease from 100 to 90% sea ice can indicate that a process is active. We have added elements of this text to the introduction, results and discussion, and conclusion.

2. Lines 200-206 & Fig. 6-7: I would be hesitant to agree with the polynyas detected along the MIZ in the Atlantic Sector, and would recommend more discussions on if these are false positives, and if similar polynya detections are common in the time series. Also, I feel Figs. 6-7 do not convey enough information to do justice to the authors' analyses. For example, the authors could present a monthly time series of a cropped-in region of interest (e.g. an area of known polynya formation) in a given year in tightly-packed sub-figures, or a time series for a given month/date for all the years.

In response to your comment as well as Reviewer 1's second major comment, we have added a time series analysis of the modelled polynya activity in the Laptev and Kara seas, two hotspots of polynya activity in "the real world", and compare the values obtained with our method to those with different fixed SIC thresholds. We do not compare these values to observations in the same regions since the model -like all models- has biases and their analysis is beyond the scope of this paper. In fact, it is the topic of an ongoing dedicated study led by our group. The new analysis for this manuscript includes a new figure, Figure 8, and Appendix Table A2. We find a strong correlation often exceeding 0.9 regardless of the diagnostic, region, or SIC threshold we compare it to, showing that our method captures very well the variability in modelled polynya activity in the region. Our polynya areas often fall between those of the 50 and 60% thresholds, but the offset is not constant, indicating again that our method is comparatively flexible

3. It would be good to know how the method transfers to the AMSR-E based 12.5km SIC product by NSIDC. If re-running on some of these data is not within the scope of the paper, some mentioning of this in the Discussions would be helpful for the readers to understand the potential of your methods combined with this SIC product with improved spatial resolution.

Re-running is indeed beyond the scope of the paper, not least because we ran out of computing time on the national server doing the CMIP6 study mentioned in the previous response. But as we tested on the range of CMIP6 models, moving the method to a different grid is no problem at all for the GMM, except that the computing time increases rapidly with spatial resolution. For our purpose, it is faster to apply our pre-trained Unet to data interpolated onto the 25km grid than to re-train Unet onto the new grid, but such re-training would be straightforward: only the line specifying the input/output spatial dimension would need to be changed. We have added a few sentences explaining this to the end of section 3.1.

Minor comments

Line 4: 'from 0.1 to 2.5%' could be confusing. Perhaps 'from 10% to 2.5%.'

No, we meant 0.1%. We have added the percent sign for clarification.

Line 16: 'not begin until' is a bit awkward. Consider rewording.

We assume the author meant L26. We rephrased.

Line 40: "'zooms' on the image": I suppose you mean "'zooms in' on the image"

Rephrased

Line 131: what are the considerations going into keeping a 'land' class rather than directly applying a land mask and classifying pack ice vs open ocean? Could using a land mask help solve the issues encountered in Lines 132-133 and in Lines 147-149?

We do not keep a land class: we prompt the GMM to split the image into a series of classes, and it automatically returns one that contains the land, regardless of the number of classes. The GMM cannot work with NaNs, and for subsequent analysis we do not want to work with 1D position arrays where the land points have been removed, so keeping known land points was the easiest. The challenge is anyway not to get rid of the land, which as you say we can easily filter out using a land mask. The issue is to separate the MIZ from polynyas.

Line 144 & in Fig. 3 caption: randomly affected -> randomly assigned?

Rephrased

Line 157: consider moving '(Table 1, first column)' to after 'The performances' to avoid confusion.

Rephrased

Fig. 4 caption: '2023 stops in April' -> '2023 data stops in April.' 'higher sea ice values' -> 'higher values in sea ice area and extent'

Rephrased

Line 180: 'a polynya' -> 'polynyas'

Corrected

Line 180: 'strongly decreases compared to' could be confusing which leads the reader to think of the time dimension. Consider changing to e.g. 'is significantly less than'

Rephrased as suggested

Line 181: the sentence after 'is not perfect' is confusing to read. Consider rewording. Do you mean 'the MIZ mask filters out some polynya pixels whose surrounding ice also has relatively low SIC, while in some complex MIZ regions, the MIZ mask fails to filter out some false positives?'

Rephrased as suggested

Line 203: 'we do not have any truthing data for it' – if not truthing, perhaps some comparisons with observation-based polynya detections in Section 3.1?

See response to your and Reviewer 1's second major comments: we have modified this part and included a quantified analysis.

Line 204: 'the occasional false positive pixel survives' -> 'occasional false positive pixels survive'

Original phrasing kept – copy editors can decide

Line 214: 'The one inconvenient' -> 'One inconvenience'

Rephrased as suggested

Line 234: 'no false negative' -> 'no false negatives'

Corrected

Throughout the manuscript:

'Timeseries' should be changed to 'time series'

Corrected

After the first usage of 'SIC', avoid using 'sea ice concentration' to be consistent.

Disagreed – most people will only read part of the manuscript and/or be tired while reading. It does not hurt to spell the acronym out once per section.

---

## Referee Report (RR1)

**Review on the revised version of "Automatic detection of Arctic polynyas using hybrid supervised-unsupervised deep learning" by Céline Heuzé and Carmen Hau Man Wong** // submitted to The Cryosphere

The revised version of the submitted manuscript has addressed all the major points I raised in round 1 (as well as those of reviewer #2) and, where necessary, implemented appropriate changes. These most prominently concern some initial gaps with regard to a long-term characterisation of the results and an assessment of seasonal effects in the approach to detecting polynyas. To this end, new, high-quality figures have been added and some previous figures have been moved to the appendix. Overall, this has noticeably improved the manuscript as a whole.  In its current form, I see no major obstacles that could prevent its swift publication (other than potential minor technical issues such as notations, etc., in the course of typesetting).

Just two final bonus questions out of curiosity from my side:

(1) A question that came up only after my initial review when rereading e.g. line 32 ("In these regions, they then use a fixed threshold in sea ice properties to distinguish polynyas from the pack ice"): I did not find any mention of another old but rather frequently used polynya detection method – the Polynya Signature Simulation Method ("PSSM", Markus and Burns (1995)). Any thoughts or even experience on how this approach would compare to your results, and/or how it could even be utilized to some extent in future ML approaches?

(2) Speaking of future ML approaches for automatic polynya detection approaches and your own depiction of "paving the way" for these: In what areas do you expect to see major methodical advancements in the next few years for A) your presented low-computational-cost approach and B) a hypothetical high-computational approach that fully utilizes higher spatial and temporal resolutions from both observations and (km-scale) modelling?